# CLIP-Fields: Weakly Supervised Semantic Fields for Robotic Memory

Nur Muhammad (Mahi) Shafiullah[†1]    Chris Paxton[2]    Lerrel Pinto[1]    Soumith Chintala[2]    Arthur Szlam[2]

*Abstract*— We propose CLIP-Fields, an implicit scene model that can be trained with no direct human supervision. This model learns a mapping from spatial locations to semantic embedding vectors. The mapping can then be used for a variety of tasks, such as segmentation, instance identification, semantic search over space, and view localization. Most importantly, the mapping can be trained with supervision coming only from web-image and web-text trained models such as CLIP, Detic, and Sentence-BERT. When compared to baselines like Mask-RCNN, our method outperforms on few-shot instance identification or semantic segmentation on the HM3D dataset with only a fraction of the examples. Finally, we show that using CLIP-Fields as a scene memory, robots can perform semantic navigation in real-world environments. Our code and demonstrations are available here: **https://mahis.life/clip-fields**

## I. INTRODUCTION

Recently, a class of models for representing 3D scenes implicitly [1] has shown great promise as a tool for computer vision [2], [3]. These neural radiance fields (NeRFs), and implicit neural representations more generally [4], can serve as differentiable databases of spatio-temporal state that can be used by robots for scene understanding, SLAM, and planning [5]–[8].

Another line of recent work has shown that web-scale weakly-supervised vision-language models (e.g. CLIP [9]) capture powerful semantic abstractions. These have proven useful for a range of robotics applications, including object understanding [10] and multi-task learning from demonstration [11]. These applications have been limited, however, by the fact that these trained representations assume a single 2D image as input; it has been an open question how best to use these to enable 3D reasoning with all the advantages these vision-language models have to offer.

In this work, we introduce a method for building weakly supervised semantic neural fields, called CLIP-Fields. The key idea is to build a mapping from locations in space $g(x, y, z) : \mathbb{R}^3 \to \mathbb{R}^d$ that serves as a generic differentiable spatial database. The database is augmented with "modality" specific heads that interface $g$ to off-the-shelf weakly-supervised language and vision models, which are used to train $g$ and the heads. We assume that we have access to depth images of the scene of interest, and approximately, the corresponding 6D camera poses. From these, we train CLIP-Fields with a contrastive loss that penalizes mismatches

1. New York University
2. FAIR Labs
† Corresponding author, email: mahi@cs.nyu.edu

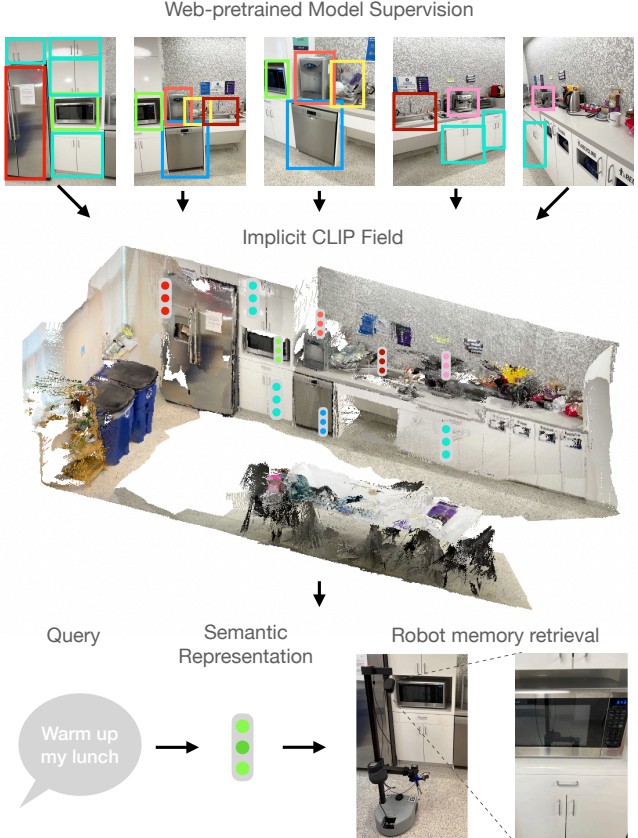

**Fig. 1:** Our approach, CLIP-Fields, integrates multiple views of a scene and can capture 3D semantics from relatively few examples. This results in a scalable 3D semantic representation that can be used to infer information about the world from relatively few examples and functions as a 3D spatial memory for a mobile robot.

between the vector output of the modality-specific head at the back-projected point in space corresponding to a pixel in an image, and the web-image-trained vectors corresponding to the location in the image; but encourages differences with vector representations of other images and regions of space.

Thus, from the point of view of a robot using CLIP-Fields as a spatial database for scene-understanding, training $g$ itself can be entirely self-supervised – the full pipeline, including training the underlying image models, need not use any explicit supervision. On the other hand, as we will show in our experiments, the spatial database $g$ can naturally incorporate scene-specific labels, if they are available.

We demonstrate our method quantitatively on instance segmentation and identification. Furthermore, we give qual-

itative examples of image-view localization, where we need to find the spatial coordinates corresponding to an image and localizing text descriptions in space. Finally, we demonstrate the use of CLIP-Fields as a differentiable geometric database for a real robot by having the robot move to look at various objects in 3D given simple natural language commands. In each of these settings, we show how our approach can encode scene-specific semantic information with very few, or even zero, human-labeled examples.

## II. RELATED WORK

Much recent progress on vision-language navigation problems such as ALFRED [12] or RXR [13] has used spatial representations or structured memory as a key component to solving the problem [14]–[17]. HLSM [15] and FiLM [14] are built as the agent moves through the environment, and rely on a fixed set of classes and a discretization of the world that is inherently limiting. In addition, these works assume a novel environment at each episode, which is not always the case – a real assistant robot might explore the same environment many times. Other representations [16] do not allow for 3D spatial queries, or rely on dense annotations, or accurate object detection and segmentation [17]–[19]. However, there is a recent trend towards using NeRF-inspired representations as the spatial knowledge base for robotic manipulation problems [6], [8].

Recent works have shown that it is possible to use supervised web image data for self-supervised learning of spatial representations. Our work is closely related to [20], where the authors show that a web-trained detection model, along with spatial consistency heuristics, can be used to annotate a 3D voxel map. That voxel map can then be used to propagate labels from one image to another. Other works, for example [21], use models specifically trained on indoor semantic segmentation to build semantic scene data-structures.

As in [2], [22]–[25], we use a mapping (parameterized by a neural network) that associates to an $(x, y, z)$ point in space a vector with semantic information. In those works, the labels are given as explicit (but perhaps sparse) human annotation, whereas, in this work, the annotation for the semantic vector are derived from weakly-supervised web image data.

Several works [10], [11] have shown how features from weakly-supervised web-image trained models like CLIP [9] can be used for robotic scene understanding. Most closely related to this work is [26], which uses CLIP embeddings to label points in a single-view 3D space via back-projection. In that work, text descriptions are associated with locations in space in a two step process. In the first step, using an ViT-CLIP attention-based relevancy extractor, a given text description is localized in a region on an image; and that region is back-projected to locations in space (via depth information). In the second step, a separately trained model decoupled from the semantics converts the back-projected points into an occupancy map. In contrast, in our work, CLIP embeddings are used to directly train an implicit map that outputs a semantic vector corresponding to each point in space. One notable consequence is that our approach

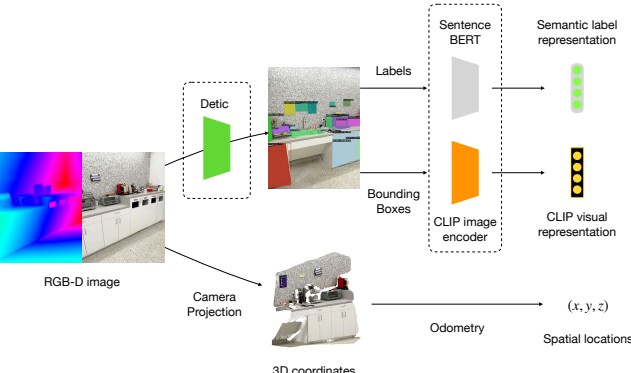

**Fig. 2:** Dataset creation process for CLIP-Fields by processing each frame of a collected RGB-D video. Models highlighted by dashed lines are off-the-shelf pre-trained models, showing that we can train a real world CLIP-Fields using no direct human supervision beyond pre-trained open label object detectors, large language models (LLMs) and visual language models (VLMs).

integrates semantic information from multiple views into the spatial memory; for example in Figure 6 we see that more views of the scene lead to better zero-shot detections.

Cohen et al. [27] looks at personalizing CLIP for specific users and rare queries, but does not build 3D spatial representations conducive to robotics applications, and instead functions on the level of individual images.

## III. APPROACH

In this section, we describe the components of our semantic scene model, and how they connect with each other.

### A. Dataset Creation

We assume access to a series of RGB-D images of a scene alongside odometry information, i.e. the approximate 6D camera poses while capturing the images. To train our model, we first preprocess such scene dataset (Fig. 2). We convert each of our depth images to pointclouds in world coordinates using the camera intrinsic and extrinsic matrices. Next, we label each of the points in the pointcloud with their possible representations. When no human annotations are available, we use web-image trained semantic segmentation models on our RGB images. We choose Detic [28] as our segmentation model since it can work with an open label set directly using CLIP embeddings. However, this model can freely be swapped out for any other pretrained detection or segmentation model. When available, we can also use semantic or instance segmentations with labels from humans.

In both cases, we derive a set of detected objects with language labels in the image, along with their label masks and potentially confidence scores. We back-project the pixels included in the the label mask to the world coordinates using our point cloud. We label each back-projected point in the world with the associated language label and label confidence score. Additionally, we label each back-projected point with the CLIP embedding of the view it was back-projected from as well as the distance between camera and the point in that particular point. Note that each point can appear multiple times in the dataset from different training images.

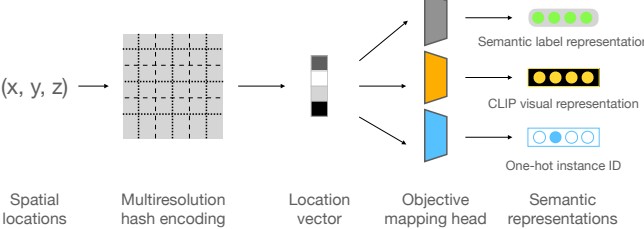

**Fig. 3:** Model architecture for CLIP-Fields. We use a Multi-resolution Hash Encoder [29] to learn a low level spatial representation mapping $\mathbb{R}^3 \to \mathbb{R}^d$, which is then mapped to higher dimensions and trained with contrastive objectives.

Thereby, we get a dataset with two sets of labels from our collected RGB-D frames and odometry information. One set of label is language-based, $D_{\text{label}} = \{(P, \text{label}_P, \text{conf}_P)\}$ where $\text{label}_P$ and $\text{conf}_P$ are just detector-given label and the confidence score to such label for each point. The second set of labels is visual, $D_{\text{image}} = \{(P, \text{clip}_P, \text{dist}_P)\}$, where $\text{clip}_P$ is the CLIP embedding of the image point $P$ was back-projected from, and $\text{dist}_P$ is the distance between $P$ and the camera in that image. We then train CLIP-Fields to combine the representations, encoding the points' semantic and visual properties in $g$.

### B. Model Architecture

Our implicit scene model can be divided into two components: a trunk $g : \mathbb{R}^3 \to \mathbb{R}^d$, which maps each location $(x, y, z)$ to a representation vector, and individual heads, one for each one of our objectives, like language or visual representation retrieval. See Figure 3 for an overview.

We parameterize $g$ as a Multi-resolution Hash Encoding as introduced in [29], with $d = 144$. We use the Multi-resolution Hash Encoding over other implicit field representations because they train relatively faster while maintaining flexibility between local and global structures, unlike purely voxel-based encodings. The objective-specific heads are simple two-layer MLPs with ReLU nonlinearities that map the 144 dimensional outputs of $g$ into higher dimensions which depend on the associated objective. These include $\text{head}_s$ that outputs a vector that matches a natural language description of what is at the point in space, and $\text{head}_v$ that matches the visual appearance of the object occupying that point in space. Optionally, we can include an instance identification head whenever we have the appropriate labels to train it.

### C. Objectives

Our implicit scene model can be simultaneously trained with multiple objectives. Each objective is trained as an implicit function that maps from real world locations in $\mathbb{R}^3$ to the objective space. We use the following objectives in training our model:

**Semantic Label Embedding:** This function encodes the semantic information of a 3D point as a $n$-dimensional representation vector. We train this using the assigned natural language labels to each point. We first convert each label to a semantic vector using a pre-trained language model trained to compare semantic similarity, such as CLIP [9] or Sentence-BERT [30]. In this paper's experiments we used Sentence-BERT for these language features, giving us $n = 768$.

**Visual Context Embedding:** This function encodes the language-aligned visual context of each point into a single vector, akin to CLIP [9]. We define the visual context of each point as a composite of the CLIP embedding of each RGB frame this point was included in, weighted by the distance from camera to the point in that frame. If it is possible to do so from the given annotation, we limit the image embedding to only encode what is in the associated object's bounding box. In this paper's experiments, we use the CLIP ViT-B/32 model, giving the visual features 512 dimensions.

**Auxilary objectives like Instance Identification**: This optional head projects the point representation to a one-hot vector identifying its instance. We use this projection head only in the cases where we have human labeled instance identification data, and the projection dimension is number of identified instances, plus one for unidentified instances.

These objectives are trained with a contrastive loss, similar to CLIP [9]. While training the contrastive loss objective, we also take into consideration the associated label weights. For the contrastive loss calculation, the loss is weighted by the semantic label confidence or negative exponential of distance from camera to point. Additionally, as is standard practice, we scale the dot product of the predicted embedding and the ground truth embedding by a learned temperature value.

Mathematically, let us assume that $P$ is the point where we are calculating the loss, $P^-$ are points with a different semantic label, $f = \text{head}_s \circ g$ is the associated semantic encoding function, $\mathcal{F}$ is a pre-trained semantic language encoder, $c$ is the confidence associated with the label at $P$, and $t$ is a temperature term, then the semantic label loss is:

$$\mathrm{L}_{\mathcal{L}}(P, f(P)) = -c \log \frac{\exp\left(f(P)^T \mathcal{F}(\text{label}_P)/t\right)}{\sum_{P^-} \exp\left(f(P)^T \mathcal{F}(\text{label}_{P^-})/t\right)}$$

Similarly, given CLIP visual embedding $C$s associated with the points, the mapping $h = \text{head}_v \circ g$, and the distance between camera and the positive point $d_P$, the visual context loss $\mathrm{L}_{\mathrm{C}}$, is:

$$\mathrm{L}_{\mathrm{C}}(P, h(P)) = -e^{-d_P} \log \frac{\exp\left(h(P)^T C_P/t\right)}{\sum_{P^-} \exp\left(h(P)^T C_{P^-}/t\right)},$$

Finally, instance identification one-hot vectors are trained with a simple cross-entropy loss $L_I$.

Then, the final loss for CLIP-Fields becomes

$$L = \alpha \mathrm{L}_{\mathcal{L}} + \beta \mathrm{L}_{\mathrm{C}} + \gamma L_I$$

where $\alpha, \beta, \gamma$ are normalizing constants to bring the loss values to a comparable scale.

### D. Training

Our models are trained with the datasets described in Sec. III-A. We train the projectors simultaneously with the contrastive losses described in Sec. III-C. Under this loss, each embedding is pushed closer to positive labels and further away from negative labels. For the label embedding

head, the positive example is the semantic embedding of the label associated with that point, while negative examples are semantic embeddings of any other labels. For the visual context embedding head, the positive examples are the embeddings of all images or image segments that contain the point under consideration, while the negative examples are embeddings of images that do not contain that point. Similar to CLIP [9], we also note that a larger batch size helps reduce the variance in the contrastive loss function. We use a batch size of $12,544$ everywhere since that is the maximum batch size we could fit in our VRAM.

## IV. EXPERIMENTAL EVALUATION

We evaluate CLIP-Fields in terms of instance and semantic segmentation in images first – to show that given good data, it can learn meaningful scene representations. Then, we show that, only using weak web-model supervision, CLIP-Fields can be used as a robot's semantic-spatial memory. Our visual segmentation experiments are performed on a subset of Habitat-Matterport 3D Semantic (HM3D semantics) [31] dataset, while our robot experiments were performed on a Hello Robot Stretch using Hector SLAM [32]. We chose HM3D semantics as our sim testing ground since the semantic labels in this dataset comes from an ad-hoc open set per scene rather than a fixed set of labels.

### A. Instance and semantic segmentation in scene images

The first task that we evaluate our model on is learning instance and semantic segmentation of 3D environments. We assume that we have access to a scene, a collection of RGB-D images in it from different viewpoints, and a limited number of them are annotated either by humans, or by a model. We consider two cases in this scenario: one where there are some human annotation data available, and in another where we are completely reliant on large, web-image trained models.

*Baselines:* In all these segmentation tasks, we use 2D RGB based segmentation models as our baselines. In all of the few-shot segmentation experiments, we take a pre-trained Mask-RCNN model with a ResNet50 FPN backbone, and a pre-trained DeepLabV3 model with a ResNet50 backbone. We fine-tune these models on each of our limited datasets, and then evaluate them on the held-out set. For the RN50 FPN model, we report the mAP at [0.5-0.95] IoU range.

*Evaluating CLIP-Fields:* Since CLIP-Fields defines a function that maps from 3D coordinates, rather than from pixels, to representation vectors, to evaluate this model's learned representations we also have to use the depth and odometry information associated with the image. To get semantic or instance segmentation, we take the depth image, using the camera matrix and odometry project it back to world coordinates, and then query the associated points in world coordinate from CLIP-Fields to retrieve the associated representations with the points. These representations can once again be projected back into the camera frame to reconstruct the segmentation map predicted by CLIP-Fields. Back-projecting to 3D world coordinates also lets CLIP-Fields correctly identify visually occluded and obstructed instances in images, which is not easy for RGB-only models.

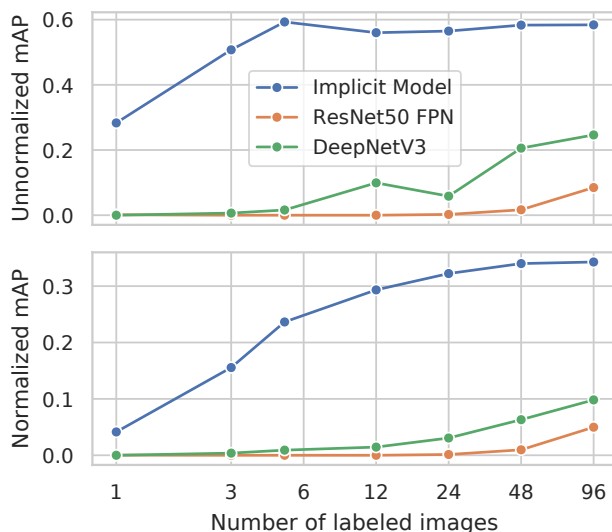

**Fig. 4:** Mean average precision in instance segmentation on the Habitat-Matterport 3D (HM3D) Semantic dataset, (top) calculated over only seen instances, and (bottom) calculated over all instances.

*1) Low-shot instance identification:* In this setting, we assume that we have access to a few images densely annotated with an instance segmentation with associated instance IDs. Such annotations are difficult for a human to provide, and thus it is crucial in this setting to perform well with very few (1-5) examples.

On this setting, we train CLIP-Fields with the provided instance segmented RGB-D images and the associated odometry data, and compare with the baseline pretrained 2D RGB models fine-tuned on the same data.

As we can see in Figure 4, the average precision of the predictions retrieved from CLIP-Fields largely outperforms the RGB-models. This statement holds true whether we normalize by the number of seen instances in the training set or by the total number of instances in the scene.

*2) Low-shot semantic segmentation:* Next, we focus on a similar setting on semantically segmenting the views from the scene from a few annotations.

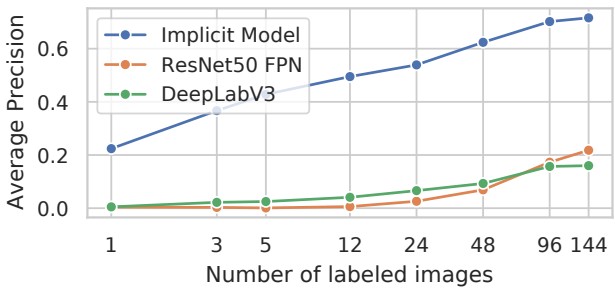

**Fig. 5:** Mean average precision in semantic segmentation on the Habitat-Matterport 3D (HM3D) Semantic dataset. Here, the average precision numbers are averaged over all semantic classes.

In Figure 5, we see once again that CLIP-Fields outperforms the RGB-based models significantly, to the point where even with three labelled views, CLIP-Fields has a higher AP than any of the baseline RGB models.

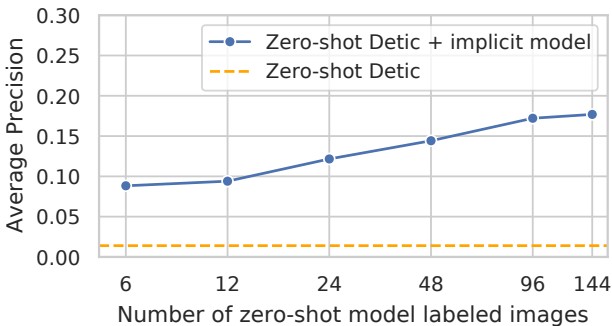

**Fig. 6:** Mean average precision in zero-shot semantic segmentation on the Habitat-Matterport 3D (HM3D) Semantic dataset.

*3) Zero-shot semantic segmentation:* To examine the benefits derived purely from imposing multi-view consistency and a 3D structure over 2D model predictions, we experiment with CLIP-Fields trained solely with labels from large web-image trained models in a zero-shot settings. In this experiment, we train CLIP-Fields only with labels given to us by such large web models, namely Detic [28]. We get the labels by using Detic on the unlabeled training images, and then train CLIP-Fields on it. Besides text labels from Detic, we also use the CLIP visual representations to augment the implicit model, as described in Section III-C.

As a baseline, we compare the trained CLIP-Fields with performance of the same Detic model used to label the scene images. Both CLIP-Fields and the baseline had access to the list of semantic labels in each scene with no extra annotations. We see in Figure 6 that enforcing 3D structure and multi-view consistency in our segmentation predictions improves the test-time predictions considerably.

In all our visual segmentation experiments, we see that enforcing 3D consistency and structure using CLIP-Fields helps identifying scene properties from images. Back-projecting the rays can also help CLIP-Fields correctly identify objects which are occluded and partially visible. This property can be extremely helpful in a busy indoor setting where not every object can be visible from every angle. Ability to work with occluded views and partial information can be a strong advantage for any embodied intelligent agent.

*4) View Localization:* Since CLIP-Fields is trained with CLIP embeddings at each coordinate, we can use such embeddings to localize an arbitrary view from the scene. To do so, we simply find the CLIP embedding of the query image. Then, we query the visual representation of the points in the scene, and take the dot product between the query representation and the point representations. Due to the contrastive loss that CLIP was trained with, points that have similar representations to the query CLIP embedding will have the highest dot product. We can use this principle to localize any view in the scene, as seen in Figure 7.

### B. Semantic Navigation on Robot with CLIP-Fields as Semantic-Spatial Memory

Training a CLIP-Fields with available data, whether they are labeled by humans or pretrained models, gives us a map-

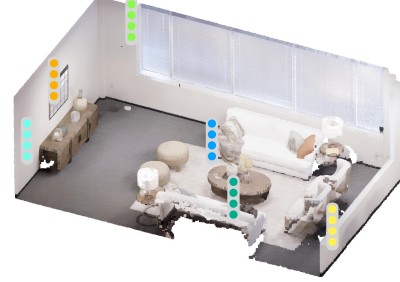

Trained CLIP-Field

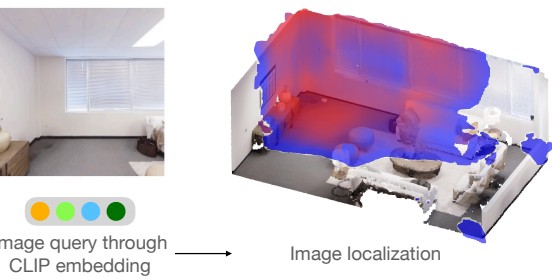

Image query through CLIP embedding → Image localization

**Fig. 7:** View localization using a trained CLIP-Fields. We encode the query image on the bottom left to its CLIP representation, and visualize the locations whose CLIP-Fields representations have the highest (more red) dot product with the embedded image. Lower dot products are blue; and below a threshold are uncolored.

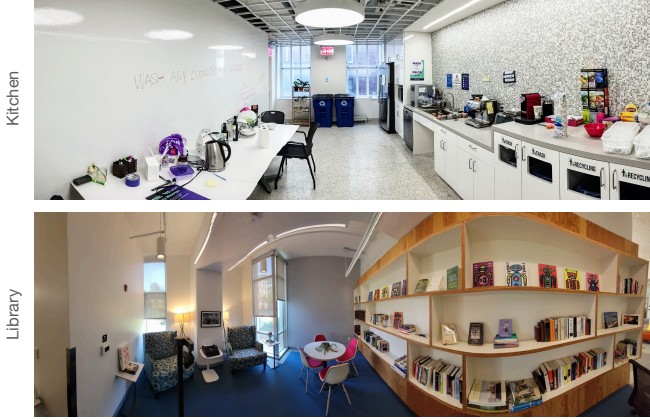

**Fig. 8:** Scenes for our real-world semantic navigation experiments. The top scene is a lab kitchen and the bottom is a library/lounge.

ping from real world coordinates to a vector representation that is trained to contain their semantic and visual properties (Section III-C). In this section, we evaluate the quality of the learned representations by using the learned model for downstream robot semantic navigation tasks.

*1) Task setup::* We define our robot task in a 3D environment as a "Go and look at *X*" task, where *X* is a natural language query defined by the user. To test CLIP-Fields's semantic understanding capabilities, we formulate the queries in two different categories:

- *Literal queries:* At this level, we choose *X* to be the literal and unambiguous name of an object present in the scene, such as "the refrigerator" or "the typewriter".
- *Visual queries:* At this level, we add references to objects by their visual properties, such as "the red fruit

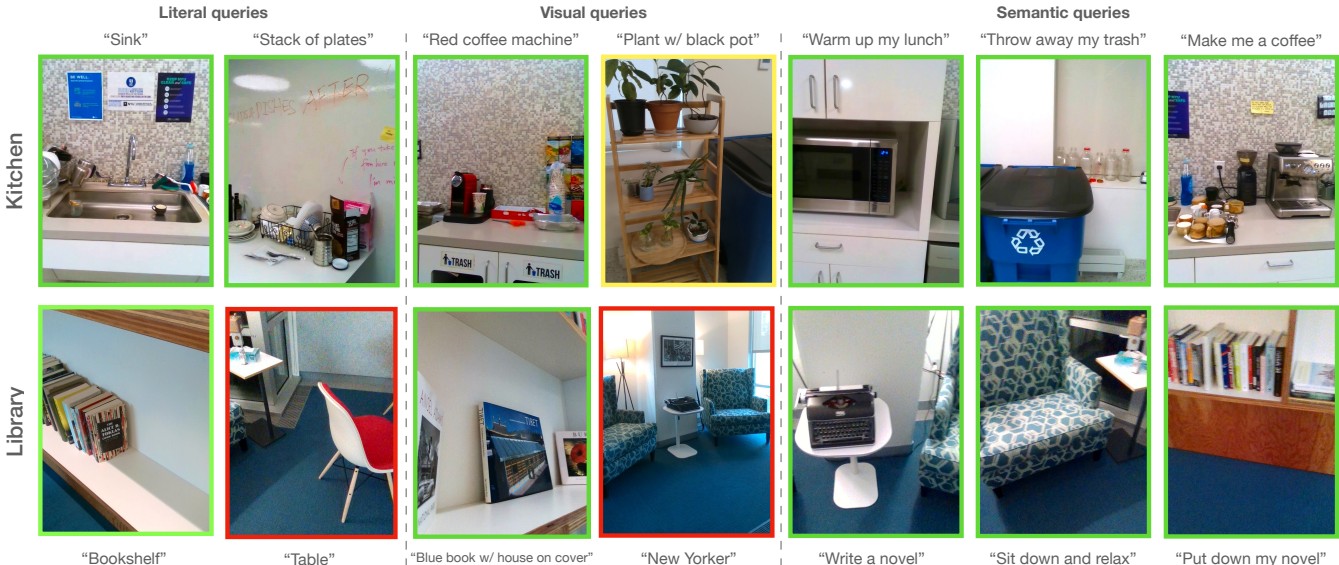

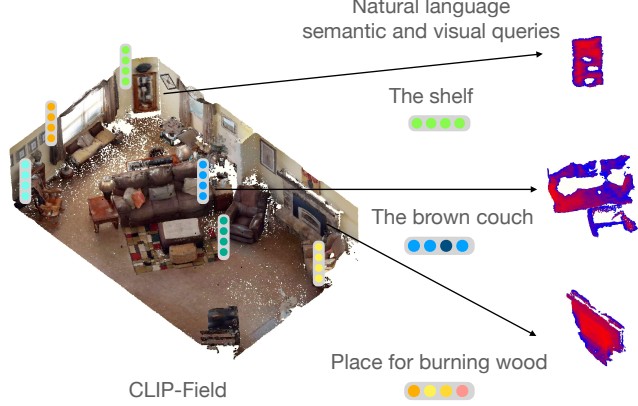

**Fig. 9:** Examples of the robot's semantic navigation in two different testing environments, looking at objects given different queries. The images show the robot's POV given the associated query, with color coded borders showing approximate correctness. The rows show different two different scenes, top being in a lab kitchen and the bottom in our lab's library/lounge space, shown in detail in figure 8.

**Fig. 10:** Running semantic queries against a trained CLIP-Fields. We encode our queries with language encoders, and compare the encoded representation with the stored representation in CLIP-Fields to then extract the best matches.

bowl" or "the blue book with a house on the cover".

- *Semantic queries:* At this level, we add references to objects by their semantic properties, such as "warm my lunch" (microwave), or "something to read" (a book).

*2) Data collection and training:* We ran our robot experiment in two different scenes, one in the lab kitchen, and another in the lab library (figure 8). For each of the scenes, we collected the RGB-D and odometry data with an iPhone 13 Pro with LiDAR sensors. The iPhone recording gave us a sequence of RGB-D images as well as the approximate camera poses in real world coordinate. On each of these scenes, we labelled a subset of the collected RGB images with Detic [28] model using ScanNet200 [33] labels. Then, we projected the depth images to find the associated world coordinate for each pixel, and created the training dataset with the 3D world coordinates and their associated semantic and visual features. On this dataset, we trained a CLIP-Fields to synthesize all the views and their associated labels.

*3) Robot execution:* Next, on our robot, we load the CLIP-Field to help with the localization and navigation of the robot. When the robot gets a new text query, we first convert it to a representation vector. We use Sentence-BERT to retrieve the semantic part of the query representation and CLIP text model to retrieve the vision-aligned part of the query representation. Then, we compare the representations with the representations of the XYZ coordinates in the scene to find the location in space maximizing their similarity. We optimize the dot product between the query representation and 3D points' semantic and visual representations to find the region where the dot product is maximum. We use the robot's navigation stack to navigate to that region, and point the robot camera to an XYZ coordinate where the dot product was highest. We consider the navigation task successful if the robot can navigate to and point the camera at an object that satisfies the query. We run about fifteen individual queries in each environment.

*4) Experiment results:* In our experiments (figure 10), we see that CLIP-Fields let the robot navigate to different points in the environment from semantic natural language queries. We generally observe that if an object was correctly identified by the web-image models during data preparation, when queried literally CLIP-Fields can easily understand and navigate to it, even with intentional misspellings in the query. However, if an object was misidentified during data preparation, CLIP-Fields fails to correctly identify it as well. For example, in row two, column two of figure 10, the part of the floor that is identified as a "table" was identified as a "table" by our web-image model earlier. For semantic queries, CLIP-Fields sometimes confuses two related concepts; for example, it retrieves the dishwasher for both "place to wash my hand" and "place to wash my dishes". Finally, the visual queries sometimes put a higher weight on the semantic match rather than visual match, such

as retrieving a white fruit bowl for "red fruit bowl" instead of the red bowl in the scene. However, the right object is retrieved if we query for "red plastic bowl".

## V. CONCLUSIONS AND FUTURE WORK

We showed that CLIP-Fields can learn 3D semantic scene representations from little or no labeled data, relying on weakly-supervised web-image trained models, and that we can use these model in order to perform a simple "look-at" task. CLIP-Fields allow us to answer queries of varying levels of complexity. We expect this kind of 3D representation to be generally useful for robotics. For example, it may be enriched with affordances for planning; the geometric database can be readily combined with end-to-end differentiable planners. In future work, we also hope to explore models that share parameters across scenes, and can handle dynamic scenes and objects.

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
