# OpenReview forum: "CLIP-Fields: Weakly Supervised Semantic Fields for Robotic Memory"
_robot-learning.org/CoRL/2022/Workshop/LangRob — LangRob 2022 Spotlight_

### Official Review · Reviewer_Sa5P · 2022-11-08
**Great paper**

**Rating:** 9
**Confidence:** 5

**Review:**

The paper presents a  novel approach for building weakly supervised semantic neural fields, called CLIP-Fields. In a nutshell, it learns mappings between spatial locations and semantic features, which can be leveraged for different tasks, such as view localization. The writing is concise and the method and the evaluation solid.

I think the idea of using VLMs and LMs as supervision (rather than projecting them) is quite unique, and an interesting way to ensemble outputs from multiple models. The ability to benefit from multiple models itself is also quite nice. There a few concurrent papers that try to solve a similar problem of anchoring VLMs into a spatial map representation, might be nice if the authors could briefly discuss similarities and differences to them in the related work section.

NLMap: "Open-vocabulary queryable scene representations for real world planning" Chen et al.
VLMaps: "Visual Language Maps for Robot Navigation" Huang et al.

---

### Official Review · Reviewer_TGLM · 2022-11-12
**Good paper, compelling results**

**Rating:** 9
**Confidence:** 4

**Review:**

This paper presents CLIP-Fields, a spatio-semantic model that maps 3D spatial locations to open-ended semantic concepts. Without any explicit human supervision, CLIP-Fields uses internet-data trained models like Detic, CLIP, and BERT to create dense annotations for 3D data. CLIP-Fields achieves compelling results on image segmentation tasks and also in real-world semantic navigation experiments.

Strengths:
+ No training data. CLIP-Field makes clever use of existing models like CLIP and Detic to generate annotations.
+ Compelling zero-shot segmentation and real-world navigation results.
+ Online demos are super cool.
+ Code is publically available.

Suggestions:
- A discussion of limitations and failure cases would be helpful.
- Some quantitative evaluations on the real-world navigation results would be great.
- Some details on the camera-setup (resolution etc.) for navigation would be great for reproducibility.

---

### Decision · Program_Chairs · 2022-11-15

Accept (Spotlight)